# AsyncSpade: Efficient Test-Time Scaling with Asynchronous Sparse Decoding

**Shuqing Luo** [* 1]  **Yilin Guan** [* 2]  **Pingzhi Li** [1]  **Ryan Wang** [3]  **Tianlong Chen** [1]

## Abstract

Test-time scaling (TTS) can boost LLM reasoning through long chain-of-thought (CoT), but the linear KV-cache growth amplifies the memory-bound bottleneck of LLM decoding. Query-aware sparse decoding methods can achieve state-of-the-art performance under constrained FLOP budget, but are mainly constrained by both sequential-dependent page filtering and coarse-grained token selection, hampering the serving efficiency and model performance on TTS tasks under high concurrency and long CoT scenarios, where token selection can even occupy higher runtime than the forward pipeline itself. In this paper, we first find that the query state of the current decoding token can be approximated in a unified manner from a short sliding window of recent queries, enabling training-free query-aware sparsity without sequential dependency in the decoding loop. Based on the findings, we propose **AsyncSpade**, an asynchronous framework for efficient TTS, built on two core components: (1) **a novel lightweight temporal-regressive module** that predicts the next-token query state, and (2) **an asynchronous disaggregated framework** that decouples the KV cache selection from the auto-regressive decoding loop, overlapping the token-level KV selection with the forward inference computation through asynchronism, thereby eliminating the sequential dependency without sacrificing model performance. We validate the effectiveness of AsyncSpade on common LLM serving setups with an A100 node, where AsyncSpade can fully overlap KV-cache operations with the inference pipeline within a certain workload range, **achieving theoretical optimal time-per-output-token (TPOT)**. Specifically, AsyncSpade delivers over $20\%$ reduc-

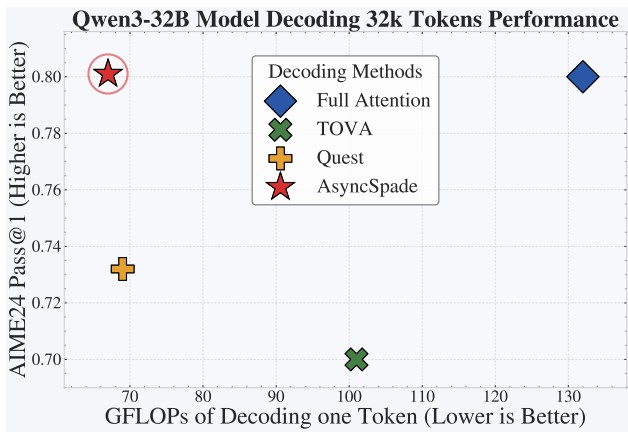

Figure 1. **Performance of Qwen3-32B on AIME24 with Long Decoding**. AsyncSpade minimizes the decoding FLOPs while maintaining high performance.

tion on TPOT compared to SoTA baseline (*i.e.* Quest) and at least $50\%$ TPOT reduction compared to full attention on Qwen3-8B and Qwen3-32B models, while matching or surpassing their accuracy on various TTS benchmarks (AIME-24/25, GPQA-Diamond, MATH-500). Our code is available through https://github.com/UNITES-Lab/AsyncSpade.

## 1. Introduction

Recent advances in large language models (LLMs) (Jaech et al., 2024; Ren et al., 2025) have demonstrated remarkable capabilities in tackling complex reasoning tasks across multiple domains, including mathematical problem solving (Ren et al., 2025; Wang et al., 2025; Huang & Yang, 2025), code generation (Ahmad et al., 2025; Huang & Yang, 2025; Guo et al., 2025), and scientific discovery (Huang et al., 2025; Wu et al., 2025), marking a pivotal advancement in the AI frontier. One of the most powerful paradigms that drives these advances is **Test-Time Scaling (TTS)**, which significantly unleashed the reasoning capabilities of LLM. Leading exemplars such as GPT-o1 (Jaech et al., 2024), DeepSeek-R1 (Guo et al., 2025), and QwQ (Team, 2024) have established that, by allocating additional computation during inference, most notably through extended chain-of-

---

[*]Equal contribution  [1]The University of North Carolina at Chapel Hill [2]Johns Hopkins University [3]Independent Researcher. Correspondence to: Tianlong Chen <tianlong@cs.unc.edu>.

*Proceedings of the 43rd International Conference on Machine Learning*, Seoul, South Korea. PMLR 306, 2026. Copyright 2026 by the author(s).

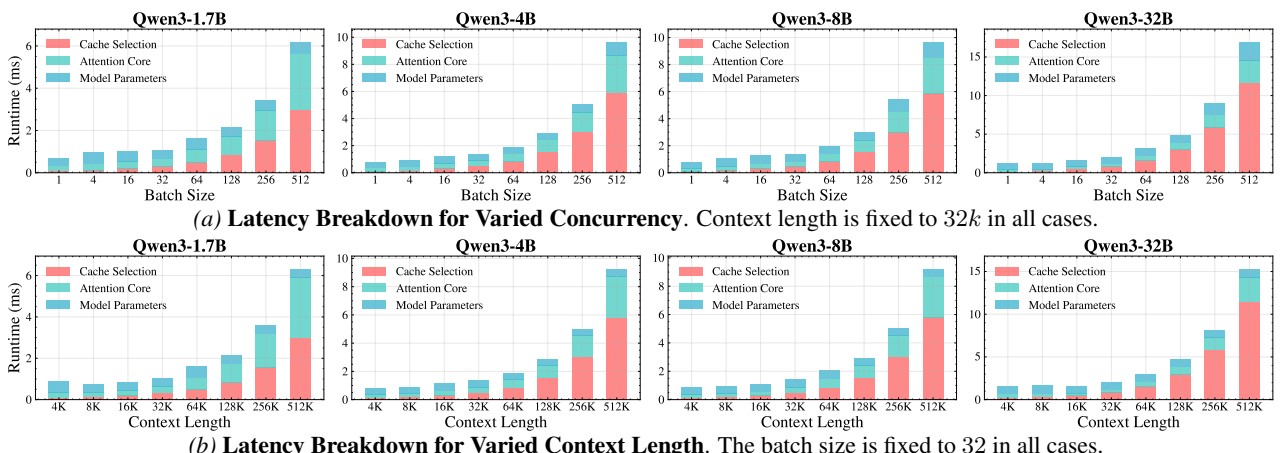

*(a)* **Latency Breakdown for Varied Concurrency**. Context length is fixed to $32k$ in all cases.

*(b)* **Latency Breakdown for Varied Context Length**. The batch size is fixed to 32 in all cases.

*Figure 2.* **Runtime Profiling for Page-level Sparse Decoding**. We benchmark the latency breakdown of a single Transformer block in the decoding stage on Qwen3 dense models (Yang et al., 2025) with an NVIDIA A100 GPU with configurations in Figure 3. We set page size to 16 following the default setting of FlashInfer (Ye et al., 2025b), and select $1/16$ tokens from the full KV cache. (a) reports results for varied batch sizes (1–512) to emulate different serving concurrency, while (b) reports results for varied context lengths ($4k$–$512k$) to emulate long chain-of-thought decoding. We further provide the corresponding latency breakdown on H200 GPUs with FlashAttention-3 in Section E, where cache selection remains a significant bottleneck on newer-generation hardware.

*Figure 3.* **Configurations of the profiled Qwen3 dense models**.

| Model | Hidden Size | # Attn Heads | # KV Heads | Intermediate Size | # Layers |
|---|---|---|---|---|---|
| Qwen3-1.7B | 2048 | 16 | 8 | 6144 | 28 |
| Qwen3-4B | 2560 | 32 | 8 | 9728 | 36 |
| Qwen3-8B | 4096 | 32 | 8 | 12288 | 36 |
| Qwen3-32B | 5120 | 64 | 8 | 25600 | 64 |

thought (CoT) (Wei et al., 2022) reasoning, TTS can unlock state-of-the-art performance on a broad spectrum of challenging tasks.

A critical challenge, however, is that TTS substantially prolongs the decoding stage. Each newly generated token must attend to the key-value (KV) cache of all previous tokens, resulting in the cost of attention computation growing linearly with the increase in decoding length. This linear expansion of the KV cache and memory footprint also intensifies the I/O pressure between GPU high bandwidth memory (HBM) and shared memory (SRAM), becoming a critical performance bottleneck and leading to exacerbated time-per-output-token (TPOT). In long-CoT reasoning tasks, this results in attention core (Dao et al., 2022), rather than the parameter computation, emerging as the dominant performance bottleneck (Sadhukhan et al., 2025), hindering the deployment of LLM in TTS scenarios under high concurrency (Agrawal et al., 2024).

One promising solution is sparse decoding, *i.e.*, approximating full attention by retaining only a small, critical fraction of tokens in the KV cache during the prolonged LLM decoding process. Previous approaches exploit fixed structural heuristics, such as preserving the "attention-sink" token (Xiao et al., 2024b) or leveraging historical patterns,

exemplified by accumulated attention scores in H2O (Zhang et al., 2023). Despite their simplicity, these query-agnostic strategies cannot fully capture factual token relevance. Recent work has highlighted that the criticality of a token strongly depends on the current query (Tang et al., 2024), leading to query-aware sparsity methods (Xiao et al., 2024a; Ribar et al., 2024; Tang et al., 2024). By directly leveraging current query embedding to dynamically select relevant KV entries, these methods can achieve superior accuracy. However, they exhibit an inherent drawback, *i.e.*, KV selection creates a sequential dependence before attention computation, as it depends on the current query state. Figure 2 demonstrates that cache selection turns out to be the dominant bottleneck of TPOT under either high concurrency or long context scenarios. Furthermore, current leading query-aware sparse attention approaches, such as Quest (Tang et al., 2024) and MoBA (Lu et al., 2025), adopt page or block-level selection strategies rather than token-level fine-grained granularity, which may ignore critical tokens and hamper model performance. This design stems from the deployment constraints. As the KV cache is stored on the same GPU for inference, modern GPU kernels such as Flash-Infer (Ye et al., 2025b) are highly optimized for reading and writing large, contiguous chunks of memory, which makes block- or page-wise access efficient. In contrast, token-level selection incurs massive, small, and irregular memory accesses that hamper runtime performance, emphasizing the necessity to advance both runtime efficiency and reasoning performance on TTS tasks.

In this paper, we propose AsyncSpade, a novel asynchronous sparse decoding framework for ultimate test-time scaling efficiency through decoupling the KV cache management & selection from the inference pipeline. To manage

these two operations separately, we introduce two specialized ranks: *Inference Rank* dedicated for forward computation, and *Cache Rank* exclusively responsible for KV management and fine-grained token selection. During inference, the *Inference Rank* asynchronously transmits query, key, and value embeddings to the *Cache Rank*. Upon receiving these embeddings, the *Cache Rank* effectively regresses the next query embedding from a sliding window of previous queries, thus enabling token-level KV selection to be prepared ahead of use. The selected KV entries are then immediately transferred back to the *Inference Rank* for efficient attention computation. This dual-rank architecture brings dual benefits. First, it eliminates the sequential selection bottleneck of existing approaches. By properly batching the inter-rank communications, both the communication and the selection operation overhead can be fully pipelined with forward inference computation. Second, it unlocks the finest possible granularity through **token-level selection** by delegating token selection to a dedicated Cache GPU. This is because the associated memory reorganization overhead can be handled asynchronously and fully overlapped, without blocking the critical inference path.

In summary, our contributions are as follows:

- **Asynchronous and disaggregated design for efficient TTS**: `AsyncSpade` first parallelizes the forward inference pipeline with token-level KV cache selection, enabling theoretically optimal TPOT within the context of query-aware sparse decoding.

- **Simple and effective next-query prediction**: Based on the insights of locality and linear correlation for adjacent consecutive query states, we introduce a lightweight, temporal locality-aware query-prediction algorithm that effectively forecasts the next query state.

- **High performance on both efficiency and TTS tasks**: `AsyncSpade` achieves comparable performance compared to full attention while consistently reducing the TPOT across common LLM serving scenarios by over 20% compared to the strong baseline of Quest and over 50% compared to the full-attention baseline.

## 2. Related Works

**Test-Time Scaling** is a effective paradigm to substantially improved the reasoning ability of LLMs by allocating extra computation at inference (Zhang et al., 2025). Existing efforts mainly follow two strategies: (1) *Sequential scaling* prolongs reasoning trajectories before producing final answers, exemplified by Long-CoT (Wei et al., 2022), and widely adopted in models such as GPT-o1 (Jaech et al., 2024), DeepSeek-R1 (Guo et al., 2025), QwQ (Team, 2024), Qwen3 (Yang et al., 2025), GPT-OSS (Agarwal et al.,

2025), and LIMO (Ye et al., 2025a). (2) *Parallel scaling* instead expands the solution space via multiple generations. Multi-sample decoding (Sun et al., 2024) and self-consistency (Wang et al., 2023) instantiate this strategy by sampling diverse reasoning paths. Beyond sampling-based methods, search-based algorithms (e.g., tree search, Monte Carlo Tree Search) (Chaffin et al., 2022; Yao et al., 2023) explicitly structure reasoning into combinatorial trajectories. While effective at boosting reasoning quality, these approaches increase inference latency and memory cost, motivating complementary strategies from the system side.

**Dynamic KV Cache Sparsity** exploits context-aware strategies to approximate attention scores. H2O (Zhang et al., 2023) utilize accumulated history attention scores to select the critical tokens. FastGen (Ge et al., 2024) adaptively compresses the KV cache by profiling attention heads and evicting tokens based on contextual focus. Loki (Singhania et al., 2024) employs offline principal component analysis calibration to reduce key vector dimensions with top-k selection. However, these approaches may still prune truly critical tokens, as they approximate relevance without conditioning on the actual query at the current decoding step.

**Query-aware KV Cache Sparsity** addresses this limitation by explicitly leveraging the query states. SparQ (Ribar et al., 2024) instantiates this method by selecting top-r dimensions of the query and pruning tokens accordingly. Quest (Tang et al., 2024) exploits query-aware estimates of attention score bounds to selectively load relevant KV cache pages. Moba (Lu et al., 2025)introduces a mixture-of-experts inspired block-wise attention that dynamically routes queries to different blocks. However, these methods either suffer from sequential dependency between token selection and inference computation, which incurs non-trivial latency, or adopt block-level coarse granularity that limits accuracy. In contrast, `AsyncSpade` achieves token-level selection without sequential dependency through a novel asynchronous and disaggregated KV cache selection mechanism.

## 3. Observation

In this section, we present two key observations on query embeddings and KV cache that motivate our approach. To quantify the similarity between two query states, we use their selectivity on the KV cache as a proxy, and introduce a metric called *overlap ratio*. For two queries $q_i$ and $q_j$, we use $S(q_i)$ and $S(q_j)$ to denote their respective selected token sets, where $|S(q)|$ denotes the number of tokens in the set $S(q)$. The *overlap ratio* is then defined as

$$\mathcal{O}_{i,j} = \frac{|S(q_i) \cap S(q_j)|}{|S|} \tag{1}$$

where $|S|$ is the fixed number of selected tokens for each

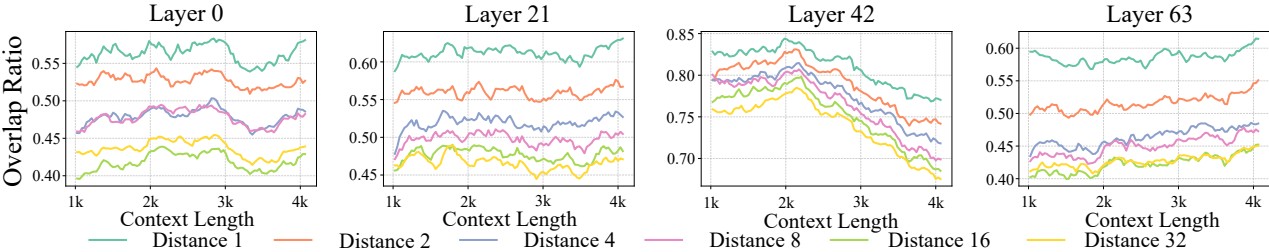

*Figure 4.* ***Overlap ratio* for query states across different token distances**. The *overlap ratio* for distance $d$ and token $t$ is examined with $\mathcal{O}_{t-d,t}$. We use Qwen3-32B and AIME24 with full attention for the profiling experiments, where the *overlap ratio* is averaged over the sample and attention head dimensions, and 4 layers are examined. $1/8$ tokens from the KV cache are selected. The per-layer statistics across all layers of DeepSeek-R1-0528-Qwen3-8B are reported in Section D, where applying the regression weights to the shifted query window consistently yields higher overlap ratio than applying them in place.

query. We assert that both $q_i$ and $q_j$ select the same number of tokens in the KV cache, *i.e.*, $|S|=|S(q_i)|=|S(q_j)|$.

### 3.1. Temporal Locality of Query States

ShadowKV (Sun et al., 2025) demonstrates that the KV cache exhibits strong temporal locality, *i.e.*, the sets of KV entries retrieved by consecutive query states share a high proportion of intersection. Inspired by this, we conduct additional profiling experiments using the proposed metric of *overlap ratio* to analyze the temporal locality of the KV cache on test-time reasoning tasks.

Figure 4 illustrates strong temporal locality in the selected KV sets, where the most recent 16 tokens consistently maintain high overlap ratios of over 40% throughout the generation process. It reveals that attention patterns of nearby queries are highly correlated, as queries at adjacent decoding steps tend to select similar KV subsets. These empirical results imply that the attention distribution of a query carries predictive information about its successors, suggesting *the feasibility of approximating the future query attention patterns by historical query information.*

### 3.2. Linear Correlation of Adjacent Queries

We further investigate whether this temporal locality can be modeled in a unified perspective and find that the historical query states possess a strong linear correlation with the current query state. We denote the sliding query states as $\{Q_{t-W}, \ldots, Q_t\}$, where $W$ is the window size. To demonstrate the linear correlation between $Q_t$ and its adjacent queries $\{Q_{t-W}, \ldots, Q_{t-1}\}$, we regress $Q_t$ from the $W$ predecessors by solving a ridge regression problem. Assuming that $Q_t$ can be expressed as a softmax-normalized combination of the windowed historical queries with raw scores $\{\omega_{t-W}, \ldots, \omega_{t-1}\}$:

$$\alpha_{t-i} = \frac{\exp(\omega_{t-i})}{\sum_{j=1}^{W} \exp(\omega_{t-j})}, \quad i = 1, \ldots, W, \quad (2)$$

$$\omega^{\star} = \arg \min_{\omega \in \mathbb{R}^W} \left\| \sum_{i=1}^{W} \alpha_{t-i} Q_{t-i} - Q_t \right\|_2^2 + \epsilon \|\omega\|_2^2. \quad (3)$$

where $\epsilon > 0$ is for regularization, $\omega_{t-i}$ is the raw regression score associated with the historical query $Q_{t-i}$, and $\alpha_{t-i}$ is its softmax-normalized counterpart used as the actual combination coefficient. The solution $\omega^{\star} = \{\omega^{\star}_{t-W}, \ldots, \omega^{\star}_{t-1}\}$ denotes the raw scores that minimize the objective above, and we write $\alpha^{\star}_{t-i} = \exp(\omega^{\star}_{t-i})/\sum_j \exp(\omega^{\star}_{t-j})$ for the corresponding normalized weights. To verify that these normalized weights faithfully reconstruct $Q_t$, we apply them back on the same historical query window:

$$\tilde{Q}_t = \sum_{i=1}^{W} \alpha^{\star}_{t-i} \cdot Q_{t-i}. \quad (4)$$

We then compare the attention score distribution induced by $\tilde{Q}_t$ and the ground-truth $Q_t$ by visualizing the *overlap ratio* of their top-$k$ token selection within the full KV cache, as demonstrated in the green line in Figure 5. The relatively high *overlap ratio* indicates that the attention pattern of $Q_t$ can be well modeled by a linear combination of its consecutive preceding query states.

## 4. Methodology

In this section, we introduce `AsyncSpade`, an algorithm-system co-design approach that optimizes TPOT for serving LLM on test-time scaling tasks through rank disaggregation, asynchronism, and fine-grained sparsity. Section 4.1 introduces the overall design principles of `AsyncSpade`, including the proposed disaggregated architecture and the workflow. Section 4.2 describes the algorithmic design for asynchronous cache selection with token-level granularity. Section 4.3 further provides the implementation details that support the ultimate decoding efficiency of `AsyncSpade`.

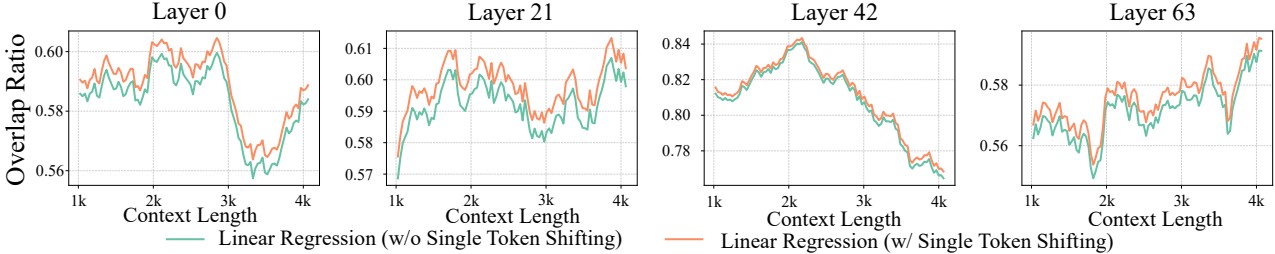

*Figure 5.* ***Overlap ratio* for linear regression w/ & w/o single-token shifting.** We follow the same settings as Figure 4. Given window size $W$, the *overlap ratio* of token $t$ for linear regression w/o single token shifting is examined by first regressing token $t$ with token $\{t - W, \ldots, t - 1\}$ and then applying the solved weights also on token $\{t - W, \ldots, t - 1\}$, while the *overlap ratio* of token $t$ w/ single token shifting is examined by first regressing token $t - 1$ with token $\{t - W - 1, \ldots, t - 2\}$ and then applying the solved weights on token $\{t - W, \ldots, t - 1\}$. $W = 16$ is used for profiling.

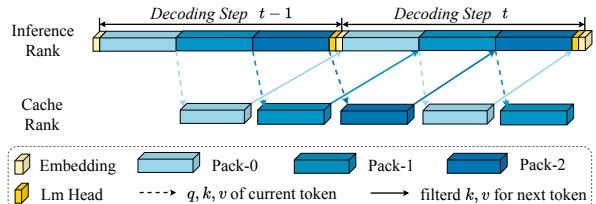

*Figure 6.* **Workflow of `AsyncSpade`.** We illustrate the overall workflow of `AsyncSpade` with 2 consecutive decoding steps, where LLM parameters are assembled into 3 packs, with fully overlapped cross-device communication and cache management, delivering theoretically optimal TPOT.

## 4.1. Asynchronous and Disaggregated Framework for Sparse Decoding

In conventional query-aware sparse decoding methods, KV selection must wait for the current query embedding to be computed, and the attention core can only be launched after the selected KV is obtained. `AsyncSpade` first breaks this dependency by decoupling the KV selection operation from the inference pipeline, moving the KV selection operation off the critical path. To achieve this, `AsyncSpade` predicts the query state of the next token and selects KV entries at token-level granularity based on it. This process can prepare the most relevant KV candidates for the next decoding step in advance, and is conducted in parallel with the inference computation pipeline.

***Inference Rank* and *Cache Rank*** To fully overlap the KV cache selection operations, we employ two specialized logical ranks to handle separate operations: *Inference Rank* for the forward inference pipeline, including attention core and parameter computation, and *Cache Rank* for managing and selecting the KV caches, coordinated through P2P asynchronous communication for efficient LLM decoding. *Inference Rank* transmits the computed query, key, and value states to the *Cache Rank* for management and selection, while *Cache Rank* returns the selected KV cache to the *Inference Rank*, enabling parallelized execution and fine-grained cache granularity.

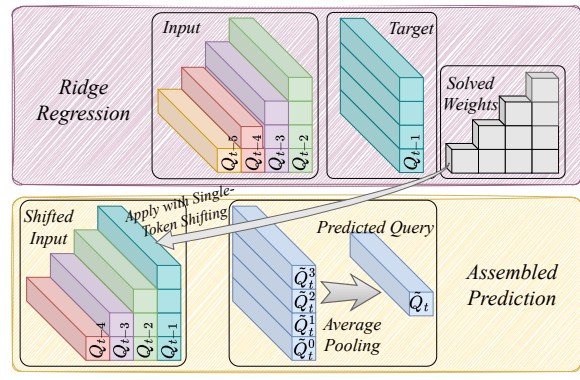

*Figure 7.* **Assembled Regression with Average Pooling in `AsyncSpade`.** We use ridge regression to learn from each historical query window. Here we set the window size to 4 for illustration.

**Communication and Computation Workflow** During LLM inference, once the generation length exceeds a predefined threshold (denoted as step $\theta_t$), the model transitions to sparse decoding mode. At step $\theta_t - 1$, immediately after computing the query state, the *Inference Rank* transfers all previous KV entries along with several recent query embeddings to the *Cache Rank* for initialization. While the *Inference Rank* continues processing forward inference for the token at step $\theta_t - 1$, the *Cache Rank* operates in parallel to predict the query embedding for the next step $\theta_t$ using a sliding window over recent queries, and applies this predicted query to select the KV cache. The selected KV pairs are subsequently transferred back to the *Inference Rank* and participate in the attention computation of the decoding step $\theta_t$. For the subsequent decoding steps, the query & key & value states of the current step are packed and passed to the *Cache Rank*. The *Cache Rank* appends KV states to the KV cache and enqueues the query state to the sliding window, then proactively performs **token-level KV cache selection** required for the next decoding step.

## 4.2. Criticality Estimation with Historical Sliding Query

A central challenge in sparse decoding is identifying the most critical set of key-value (KV) pairs to maintain model

performance. Conventional query-aware methods (Tang et al., 2024; Lu et al., 2025) directly use the query state at the current decoding step for cache selection to adhere to the definition of attention core (Vaswani et al., 2017). However, this sequential dependency and coarse-grained granularity can hamper both LLM serving efficiency under heavy concurrency and the model reasoning performance on TTS tasks (Łańcucki et al., 2025). To tackle these drawbacks, `AsyncSpade` effectively predicts the query in advance to approximate the token criticality.

**Temporal-Regressive Prediction with Adjacent Query States** Based on the insights in Section 3, `AsyncSpade` exploits the observation that consecutive query states exhibit strong temporal locality and can be well approximated as a linear combination of their predecessors. Building on these observations, we directly apply the solved normalized weights to a single-token-shifted query window to predict the query state of the next token. At current decoding step $t$, we first obtain the softmax-normalized weights $\{\alpha^\star_{t-W}, \ldots, \alpha^\star_{t-1}\}$ by solving the ridge regression problem in Equation (3) and applying the normalization in Equation (2). Each $\alpha^\star_{t-i}$ corresponds to the historical query $Q_{t-i}$ and reflects its contribution to reconstructing $Q_t$. We then apply these *same* normalized weights to the shifted query window $\{Q_{t-W+1}, \ldots, Q_t\}$—i.e., the original window advanced by one token—to *predict* the next query state $\hat{Q}_{t+1}$:

$$\hat{Q}_{t+1} = \sum_{i=1}^{W} \alpha^\star_{t-i} Q_{t+1-i}. \tag{5}$$

Figure 5 demonstrates the practicality of this strategy. The shifted prediction $\hat{Q}_{t+1}$ achieves an *overlap ratio* comparable to—and at most layers even higher than—the in-place reconstruction $\tilde{Q}_t$ in Equation (4). This is somewhat counterintuitive since $\tilde{Q}_t$ has direct access to $Q_t$ in the regression objective, yet shifting performs comparably or better in practice. We attribute this to the fact that the shifted setting more closely matches deployment: at inference, the target query $Q_{t+1}$ is genuinely unseen, and weights solved against the most recently observed query generalize better to the next step than weights overfit to $Q_t$ itself.

**Assembled Regression with Average Pooling** To better integrate the historical queries within a certain temporal range in a unified manner, we further extend the ridge regression by ensembling the predictions from multiple window sizes. We refer to this scheme as *assembled regression*, since it assembles a family of single-window predictors into a single estimate via average pooling. Rather than relying on a single window size, we perform regression across all window sizes $k \in \{1, \ldots, W\}$. For each window size $k$, we use the states $\{Q_{t-k}, \ldots, Q_{t-1}\}$ to regress $Q_t$ by first solving the ridge regression problem

defined in Equation (3) to obtain raw scores $\omega^{\star,(k)}$, followed by the softmax normalization in Equation (2) to yield the normalized weights $\mathcal{A}_k = \{\alpha^{(k)}_{t-k}, \ldots, \alpha^{(k)}_{t-1}\}$ that are actually used for prediction. Each $\alpha^{(k)}_{t-i}$ corresponds to the historical query $Q_{t-i}$. These weights are then applied to the shifted sequence $\{Q_{t-k+1}, \ldots, Q_t\}$ and yield a candidate estimation of the next query embedding $\hat{Q}^{(k)}_{t+1} = \sum_{i=1}^{k} \alpha^{(k)}_{t-i} Q_{t+1-i}$. This process generates $W$ complementary estimates $\{\hat{Q}^{(1)}_{t+1}, \ldots, \hat{Q}^{(W)}_{t+1}\}$, each capturing temporal locality at a different scale. Finally, these estimates are aggregated through average pooling to obtain the final estimation of the next query embedding:

$$\hat{Q}_{t+1} = \frac{1}{W} \sum_{k=1}^{W} \hat{Q}^{(k)}_{t+1}. \tag{6}$$

This lightweight assembled regression incurs negligible runtime on GPUs.

### 4.3. Hardware-Efficient Implementation

**Depth-wise Parallelism on Cache Rank** To fully overlap the communication overhead across *Inference & Cache Ranks* as well as better utilize the parallel computation resources on the *Cache Rank*, we assemble several consecutive transformer decoder blocks in the LLM into a packed unit to conduct asynchronous transmission between the *Inference* and *Cache Ranks* at unit granularity rather than performing separate communications for each layer. In this way, we can reduce the launch times for cross-device communication and better utilize the bandwidth. Upon receiving the bundled states from multiple blocks sent by the *Inference Rank*, the *Cache Rank* can perform selection operations with parallelism along the depth dimension.

**Batched Matmul for Different Attention Architectures** We investigate the adaptation of `AsyncSpade` on all the softmax attention variants in modern LLMs, including Multi-Head Attention (MHA (Vaswani et al., 2017)), Grouped Query Attention (GQA (Ainslie et al., 2023)), and Multi-Query Attention (MQA (Shazeer, 2019)). Specifically, the Multi-Head Latent Attention in the DeepSeek series (Liu et al., 2024a;b; Guo et al., 2025) and Kimi-K2 (Team et al., 2025) can be exactly transformed into MQA during the inference stage through matrix absorption. Here, we treat all these architectures as variants of GQA. Denoting the number of attention (query) heads as $N_q$ and the number of key & value heads as $N_{kv}$. To be specific,

- MHA contains $N_q$ query groups with $N_q = N_{kv}$.

- GQA contains $N_q/N_{kv}$ query groups, and $N_q$ should be divisible by $N_{kv}$.

- MQA/MLA contains only 1 query group with $N_{kv} = 1$.

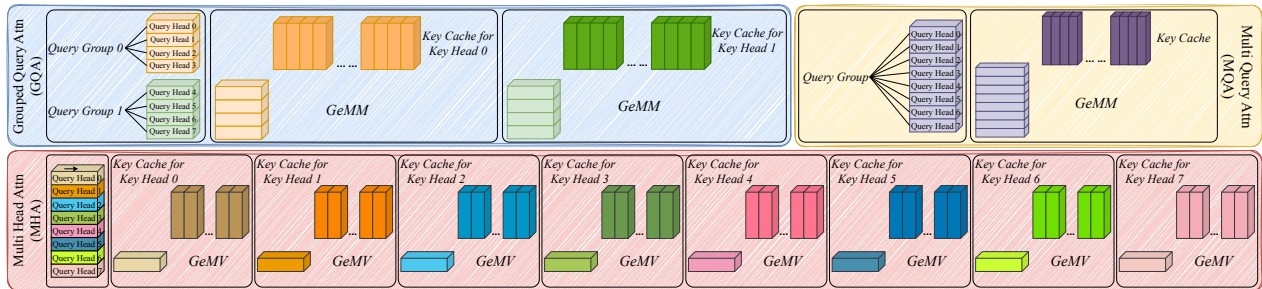

**Figure 8.** **Apply** `AsyncSpade` **to Different Attention Architectures through Batched MatMul**. While conventional Multi-Head Attention (MHA) is restricted to implementing criticality estimation only through GeMV, the Group Query Attention (GQA) and Multi-Query Attention (MQA/MLA) architectures prevalent in modern LLMs can achieve this through GeMM.

We present a comprehensive investigation on applying `AsyncSpade` to these attention variants using native Py-Torch interfaces in Figure 8. Denoting the batch size as $bs$, head dimension as $D_h$, and the number of tokens in the KV cache as $N_t$. Since the query state only contains one token during the decoding stage, the tensor shape can be formulated as $(bs, N_{kv}, N_q/N_{kv}, D_h)$, and the key states can be correspondingly formulated as $(bs, N_{kv}, N_t, D_h)$. `AsyncSpade` for MHA can only be implemented with flattened GeMV, while other architectures can be implemented with flattened GeMM, which can better utilize the tensor core, the matrix computation unit on NVIDIA GPUs, for better runtime efficiency. Since most of the modern LLMs are built on GQA or MLA, the *Cache Rank* GPUs can therefore be effectively utilized for token criticality estimation.

**Communication-Computation Overlap** We illustrate the overall workflow in Figure 6, which presents the communication-computation overlapping strategies in `AsyncSpade`. The hardware requirements for `AsyncSpade` lie in two aspects:

- The latency of a P2P communication cycle, including (1) sending the packed key & value (& query) states from *Inference Rank* to *Cache Rank*, and (2) sending the packed selected KV cache from *Cache Rank* to *Inference Rank*, should be less than the corresponding forward computing latency on the *Inference Rank*.

- The latency of processing a packed state on the *Cache Rank*, including (1) token criticality estimation, (2) top-$k$ selection, and (3) KV cache re-organization for selected tokens, should be lower than the corresponding forward computing latency on the *Inference Rank*.

**Configurable Rank Allocation** The split of GPUs between *Inference Rank* and *Cache Rank* is configurable rather than fixed. The general principle is that the *Inference Rank* hosts the model layer partitions and dictates the per-step forward latency budget, while the *Cache Rank* only needs sufficient resources to complete (1) temporal-regressive query

*Table 1.* **Hardware Specs for the 2 Nodes**.

| Node Specs | $N_1$ | $N_2$ |
| --- | --- | --- |
| GPU | 8×A100 SXM | 8×H100 SXM |
| GPU Memory | 80GB | 80GB |
| Inter-GPU Bandwidth | 250 GB/s | 350 GB/s |

prediction, (2) token-level top-$k$ selection, and (3) KV cache reorganization within the time window of one forward pass on the *Inference Rank*. The optimal allocation ratio therefore depends on the specific deployment scenario (model size, batch size, and context length).

## 5. Experiments

### 5.1. Setups

**Models, Datasets, and Baselines** We evaluate `AsyncSpade` across four popular and challenging test-time reasoning benchmarks: AIME24, AIME25 (Art of Problem Solving, 2025), GPQA-Diamond (Rein et al., 2024) and MATH500 (Hendrycks et al., 2021). We employ state-of-the-art open-source LLMs in our experiments, specifically Qwen3-32B (Yang et al., 2025) and DeepSeek-R1-0528-Qwen3-8B (Guo et al., 2025). To make a rational and comprehensive comparison, we benchmark against leading query-aware sparse attention algorithms, including TOVA (Oren et al., 2024) and Quest (Tang et al., 2024).

**Hardware Configuration and Hyperparameters** Our experiments are conducted on two nodes: $N_1$ equipped with $8\times$ NVIDIA A100 (80 GB) GPUs and $N_2$ with $8\times$ NVIDIA H100 (80 GB) GPUs. The hardware specs are provided in Table 1. $\epsilon$ is set to $1e-5$. During inference stage, we fix the temperature to $0.8$ and use top-$p$ sampling with $p$ set to $0.4$. We average the results over 8 seeds.

**Model Parallelism Configuration** In our experiments, we only use pipeline parallelism as the model parallelism scheme for inference, *i.e.*, partitioning model layers across different GPUs, and ensuring that the packed layers for P2P communication are placed on the same device.

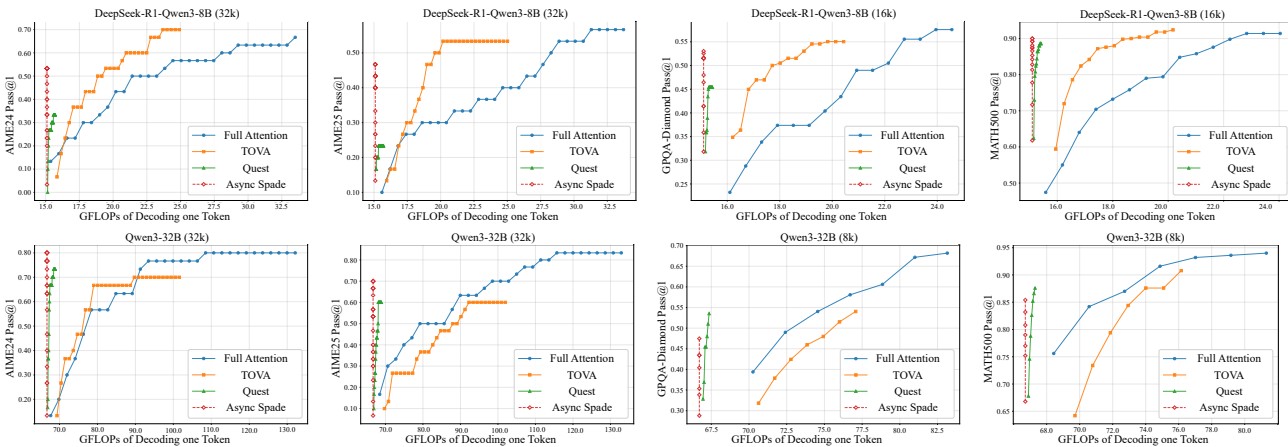

*Figure 9.* **Performance Comparison on TTS Benchmarks**. We examine the Pass@1 ($y$-axis) and the FLOPs of decoding one token ($x$-axis) for 8B and 32B models with $2k$ selected tokens for sparse methods, and `AsyncSpade` consistently outperforms the strong baselines and achieves the highest performance while consuming the least FLOP budgets.

## 5.2. Performance Comparison

We systematically evaluate the performance of `AsyncSpade` on popular test-time scaling benchmarks, using three strong baselines: Quest (Tang et al., 2024) with page size 16, TOVA (Oren et al., 2024), and vanilla full attention. We present the comprehensive performance comparison in Figure 9, where the Pass@1 solving rate is used as the $y$-axis and the FLOPs of decoding one token is used as the $x$-axis. We provide the definition details of Decoding FLOPs in Section A. We select $2k$ tokens for each sparse decoding method. `AsyncSpade` consistently outperforms other approaches with minimized FLOPs budget and comparable or better solving rate. We also evaluate the long-context retrieval capability of `AsyncSpade` on the Needle-in-a-Haystack test in Section C, where `AsyncSpade` closely matches full attention and substantially outperforms trivial query-prediction baselines.

## 5.3. Efficiency Comparison

We benchmark the TPOT performance of `AsyncSpade` and other baselines under the same decoding context length and concurrency level, visualized in Figure 10. `AsyncSpade` achieves theoretically minimized TPOT on cutting-edge data-center GPUs. To be specific, **(1) Compared to Quest**, `AsyncSpade` can fully overlap the KV cache selection overheads; meanwhile, `AsyncSpade` can also be deployed with any LLM inference backends rather than restricted to the paged inference engine (Ye et al., 2025b) in Quest. **(2) Compared to TOVA**, which offers higher accuracy than Quest but suffers from limited practicality, `AsyncSpade` makes this fine-grained strategy deployable through the innovative asynchronous and disaggregated design. Specifically, TOVA relies on se-

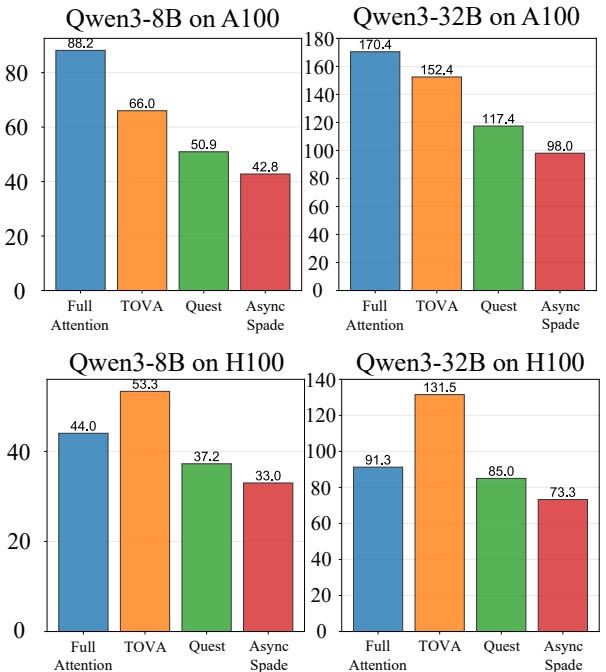

*Figure 10.* **Runtime Comparison Across Multiple Configurations**. TPOT is examined with batch size 8 on $32k$ decoding context. `AsyncSpade` consistently outperforms other strong baselines with minimized TPOT under concurrent serving and long decoding scenarios.

quential and GPU-inefficient top-$k$ selection, which incurs significant overhead that even negates latency gains from sparsity when the generation length and $k$ are large.

## 5.4. Ablation on the Latency and Bandwidth

We tests overheads of different components in `AsyncSpade`. Table 2 shows that the cache management latency is consistently lower than inference

*Table 2.* **Ablation Studies for `AsyncSpade`**. We investigate the cache management latency and inter-GPU communication bandwidth required for fully overlapping communication overheads on Qwen3-8B and 32B models, using both A100 and H100 nodes.

| | Device | #Packed Layers | Inference | Cache Operation | Minimal Bandwidth |
|---|---|---|---|---|---|
| Batch Size 8 | A100 | 6 Layers | 7.13 ms | 6.42 ms | 107.71 GB/s |
| Qwen3-8B | | 12 Layers | 14.26 ms | 12.75 ms | 107.71 GB/s |
| 32k tokens | H100 | 6 Layers | 5.47 ms | 3.92 ms | 140.40 GB/s |
| Select 2k | | 12 Layers | 10.94 ms | 7.85 ms | 140.40 GB/s |
| Batch Size 16 | A100 | 6 Layers | 7.30 ms | 7.02 ms | 105.20 GB/s |
| Qwen3-8B | | 12 Layers | 14.60 ms | 14.43 ms | 105.20 GB/s |
| 16k tokens | H100 | 6 Layers | 5.91 ms | 5.18 ms | 129.94 GB/s |
| Select 1k | | 12 Layers | 11.82 ms | 11.14 ms | 129.94 GB/s |
| Batch Size 8 | A100 | 4 Layers | 6.30 ms | 5.16 ms | 159.49 GB/s |
| Qwen3-32B | | 8 Layers | 12.60 ms | 8.32 ms | 159.49 GB/s |
| 32k tokens | H100 | 4 Layers | 4.39 ms | 3.74 ms | 228.88 GB/s |
| Select 2k | | 8 Layers | 8.78 ms | 7.48 ms | 228.88 GB/s |
| Batch Size 16 | A100 | 4 Layers | 7.77 ms | 7.02 ms | 129.32 GB/s |
| Qwen3-32B | | 8 Layers | 15.54 ms | 14.01 ms | 129.32 GB/s |
| 16k tokens | H100 | 4 Layers | 4.37 ms | 3.72 ms | 229.93 GB/s |
| Select 1k | | 8 Layers | 8.74 ms | 7.48 ms | 229.93 GB/s |

time across all configurations. We also compute the minimal bandwidth required to achieve a fully overlapped workflow for each configuration. Our analysis shows that all the bandwidth requirements can fall well within the capabilities of the corresponding hardware specifications listed in Table 1. This demonstrates that `AsyncSpade` can effectively overlap communication and KV selection overheads under practical hardware constraints.

## 6. Conclusions and Future Work

We introduce `AsyncSpade`, an algorithm-system co-design approach to optimize TPOT for LLM decoding on TTS tasks. By asynchronously disaggregating KV-cache management from the forward pass, `AsyncSpade` eliminates redundant operations in the model inference pipeline, delivering both heavy concurrency support and high performance, especially achieving theoretically optimal TPOT on common LLM serving scenarios. As a pioneering effort in training-free, high-performance, and efficient TTS, `AsyncSpade` opens avenues to further boost LLM decoding efficiency while maintaining high generation quality.

## Impact Statement

This paper aims to improve the communication efficiency for inference with modern attention-based LLMs. The efficiency advantage of such models might help democratize access of language models. On the other hand, whether such new algorithm would affect known issues such as biased and harmful outputs of language models remains an unexplored research question.

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

# A. Definition of Decoding FLOPs for one Token

**FLOPs for Different Decoding Strategies**  We provide the theoretical computation cost in terms of floating-point operations (FLOPs) for different decoding strategies, following the parameter settings of DeepSeek-R1-0528-Qwen3-8B and Qwen3-32B. The total computation cost is decomposed into two parts: (1) parameter compute cost $C_{\text{param}}$, which comes from the dense linear layers (attention projection, key/value projection, and feed-forward network), and (2) attention-related cost $C_{\text{attn}}$, which varies with the decoding strategy.

The parameter compute cost can be written as

$$C_{\text{param}} = l \cdot \big(2 \cdot 2 \cdot H \cdot q \cdot h + 2 \cdot 2 \cdot kv \cdot h \cdot H + 3 \cdot 2 \cdot H \cdot i\big), \tag{7}$$

where $l$ is the number of layers, $H$ is the hidden size, $q$ is the number of attention heads, $kv$ is the number of KV heads, $h$ is the head dimension, and $i$ is the intermediate dimension of the feed-forward network.

The attention cost depends on the sparsity strategy:

- **Full Attention**:
$$C_{\text{attn}} = l \cdot (4 \cdot q \cdot h \cdot T), \tag{8}$$
    where $T$ is the total sequence length.

- **TOVA**:
$$C_{\text{attn}} = l \cdot (4 \cdot q \cdot h \cdot C \; + \; 2 \cdot q \cdot h \cdot T), \tag{9}$$
    where $C$ is the number of selected tokens.

- **Quest**:
$$C_{\text{attn}} = l \cdot (4 \cdot q \cdot h \cdot C \; + \; 2 \cdot q \cdot h \cdot (T/P)), \tag{10}$$
    where $P$ is the page size for block-level sparsity.

- **`AsyncSpade` (Our Method)**:
$$C_{\text{attn}} = l \cdot (4 \cdot q \cdot h \cdot C). \tag{11}$$

Finally, the total FLOPs is obtained as

$$C_{\text{FLOPs}} = C_{\text{param}} + C_{\text{attn}}. \tag{12}$$

**Theoretically Optimal TPOT**  If the cache management & KV selection operations in `AsyncSpade` can be fully overlapped by the forward inference pipeline on the *Inference Rank*, then we only need to take (1) the parameter computation and (2) the attention core computation with the selected tokens into consideration, which consumes exactly fewer FLOPs than all the other counterparts, *i.e.*, achieving theoretically optimal TPOT performance.

# B. Solving Softmax-Normalized Ridge Regression

We provide the detailed pseudo code for solving the softmax-normalized ridge regression problem in `AsyncSpade` in Algorithm 1. This solving program consumes negligible runtime even under heavy concurrency because it is very lightweight.

# C. Needle-in-a-Haystack Evaluation

To further validate the effectiveness of `AsyncSpade` as a query-aware sparse decoding method and the overlap ratio as a proxy for the model performance, we conduct supplementary experiments using the Needle-in-a-Haystack (NIAH) test. Specifically, we construct a challenging evaluation by inserting a single short needle that embeds a 20-digit random code associated with a randomly sampled city, phrased in a low-salient way (*e.g.*, "*reference code . . .*") rather than with an explicit memorization cue. The needle is placed at 50 evenly spaced relative depths within the context, and each configuration is repeated for 2 rounds. We examine Qwen3-32B at varied context lengths using the decoding methods discussed in our paper, and use 1/8 sparsity for all the sparse decoding methods. For comparison, we additionally include trivial query-prediction baselines, denoted as *Distance-k*, which filter the KV cache with the query token from $k$ steps earlier.

---

**Algorithm 1** Assembled KV Cache Selection

---

1: **Inputs:**
2:   • Cached query window: $\mathbf{Q}_{\text{cache}} \in \mathbb{R}^{B \times H \times W \times D}$
3:   • Current query: $\mathbf{q}_{\text{curr}} \in \mathbb{R}^{B \times H \times 1 \times D}$
4:   • Key states: $\mathbf{K} \in \mathbb{R}^{B \times H \times L \times D}$
5:   • Regularization parameter: $\epsilon$, Window size: $W$
6: **Output:** Attention logits $\mathbf{L} \in \mathbb{R}^{B \times H \times L}$

7: **Step 1: Input Preparation**
8: $\mathbf{Q}_{\text{hist}} \leftarrow \text{reshape}(\mathbf{Q}_{\text{cache}}[:, :, :W-1, :], [B \cdot H, W-1, D])$
9: $\mathbf{q}_{\text{prev}} \leftarrow \text{reshape}(\mathbf{Q}_{\text{cache}}[:, :, -1 :, :], [B \cdot H, 1, D])$
10: **Step 2: Mask Construction**
11: $\mathbf{M} \leftarrow \text{lower\_triangular\_mask}(W) \in \{0, 1\}^{W \times W}$
12: $\mathbf{M} \leftarrow \text{broadcast}(\mathbf{M}, [B \cdot H, W, W])$
13: **Step 3: Ridge Regression**
14: $\mathbf{G} \leftarrow \mathbf{Q}_{\text{hist}} \mathbf{Q}_{\text{hist}}^{\top} + \epsilon \mathbf{I}$ {Gram matrix with regularization}
15: $\mathbf{b} \leftarrow \mathbf{Q}_{\text{hist}} \mathbf{q}_{\text{prev}}^{\top}$
16: $\mathbf{w}_{\text{base}} \leftarrow \text{softmax}(-\mathbf{G}^{-1}\mathbf{b})$ {Solve ridge regression problem}
17: **Step 4: Weight Matrix Construction**
18: $\mathbf{W}_{\text{matrix}} \leftarrow \text{broadcast}(\mathbf{w}_{\text{base}}, [B \cdot H, W, W-1])$
19: $\mathbf{W}_{\text{matrix}} \leftarrow \mathbf{W}_{\text{matrix}} \odot \mathbf{M}$ {Apply mask}
20: $\mathbf{W}_{\text{matrix}} \leftarrow \text{softmax}(\mathbf{W}_{\text{matrix}})$ {Normalize}
21: **Step 5: Query Prediction**
22: **for** $j = 1$ **to** $W$ **do**
23:     $\mathbf{Q}_j \leftarrow \mathbf{Q}_{\text{cache}}[:, :, -1-j : -1, :]$
24:     $\mathbf{w}_j \leftarrow \mathbf{W}_{\text{matrix}}[:, j, :j]$
25:     $\mathbf{q}_{\text{candidate}}^{j} \leftarrow \sum_{k=1}^{j} w_j^k \cdot \mathbf{q}_j^k$ {Shifted weighted sum}
26: **end for**
27: **Step 6: Query Average Pooling**
28: $\mathbf{q}_{\text{final}} \leftarrow \frac{1}{W} \sum_{j=1}^{W} \mathbf{q}_{\text{candidate}}^{j}$ {Average over all candidate next query}
29: **Step 7: KV Cache Selection**
30: $\mathbf{L} \leftarrow \mathbf{q}_{\text{final}} \mathbf{K}^{\top}$ {Select significant KV pairs}
31: $\mathbf{L} \leftarrow \text{reshape}(\mathbf{L}, [B, H, L])$
32: **return** $\mathbf{L}$

---

*Table 3.* NIAH retrieval accuracy on Qwen3-32B with 10k context length. *Distance-k* denotes filtering KV cache with the query state from $k$ steps earlier.

| Full Attention | AsyncSpade | Distance-1 | Distance-2 | Distance-3 | Distance-4 |
|:---:|:---:|:---:|:---:|:---:|:---:|
| 0.99 | 0.93 | 0.74 | 0.56 | 0.43 | 0.48 |

*Table 4.* NIAH retrieval accuracy on Qwen3-32B with 20k context length.

| Full Attention | AsyncSpade | Distance-1 | Distance-2 | Distance-3 | Distance-4 |
|:---:|:---:|:---:|:---:|:---:|:---:|
| 0.98 | 0.97 | 0.77 | 0.77 | 0.61 | 0.64 |

The NIAH evaluations in Tables 3 and 4 demonstrate that AsyncSpade can largely preserve the long-context retrieval capability of full attention, and significantly outperforms the trivial approaches that filter the KV cache solely with a previous query token. The ranking is consistent with the overlap ratio proxy used in the main paper, where AsyncSpade > *Distance-1* > *Distance-2* > *Distance-3* > *Distance-4*, supporting the validity of overlap ratio as a proxy for downstream task performance.

## D. Per-Layer Overlap Ratio Analysis

Due to space constraints in the main paper, we did not provide a comprehensive per-layer investigation of the overlap ratio. While the primary goal of AsyncSpade is to preserve the original model capability in a general manner rather than to dominate every individual layer, here we report the average overlap ratio of every layer of DeepSeek-R1-0528-Qwen3-8B (36 layers in total) to complement the aggregated visualization in Figure 5 of the main paper. We compare three settings: (1) *Dist-1*, which filters the KV cache with the previous query token; (2) *w/ shifting*, our assembled regression with single-token shifting (Algorithm 1); and (3) *w/o shifting*, the same regression without the single-token shifting operation.

*Table 5.* Per-layer average overlap ratio on DeepSeek-R1-0528-Qwen3-8B. "Dist-1" filters the KV cache using the previous query token. "w/ shifting" is the proposed assembled regression with single-token shifting; "w/o shifting" is the same regression without shifting.

| Layer | Dist-1 | w/ shifting | w/o shifting | Layer | Dist-1 | w/ shifting | w/o shifting |
|:---:|:---:|:---:|:---:|:---:|:---:|:---:|:---:|
| 1 | 0.5266 | 0.5681 | 0.5483 | 19 | 0.5118 | 0.5456 | 0.5296 |
| 2 | 0.5511 | 0.5866 | 0.5608 | 20 | 0.5453 | 0.5752 | 0.5517 |
| 3 | 0.5892 | 0.6240 | 0.6025 | 21 | 0.5174 | 0.5405 | 0.5141 |
| 4 | 0.5878 | 0.6151 | 0.5844 | 22 | 0.5429 | 0.5584 | 0.5281 |
| 5 | 0.5672 | 0.5922 | 0.5622 | 23 | 0.5644 | 0.5849 | 0.5556 |
| 6 | 0.5891 | 0.6096 | 0.5716 | 24 | 0.6092 | 0.6226 | 0.5961 |
| 7 | 0.5290 | 0.5551 | 0.5223 | 25 | 0.5473 | 0.5544 | 0.5194 |
| 8 | 0.5524 | 0.5578 | 0.5097 | 26 | 0.5890 | 0.6043 | 0.5717 |
| 9 | 0.5665 | 0.5839 | 0.5457 | 27 | 0.5351 | 0.5480 | 0.5140 |
| 10 | 0.4863 | 0.5140 | 0.4800 | 28 | 0.5487 | 0.5685 | 0.5372 |
| 11 | 0.6300 | 0.6562 | 0.6300 | 29 | 0.5673 | 0.5781 | 0.5389 |
| 12 | 0.5574 | 0.5904 | 0.5657 | 30 | 0.5176 | 0.5382 | 0.5057 |
| 13 | 0.4876 | 0.5181 | 0.4852 | 31 | 0.5930 | 0.6184 | 0.5919 |
| 14 | 0.4824 | 0.5083 | 0.4739 | 32 | 0.5536 | 0.5764 | 0.5455 |
| 15 | 0.5570 | 0.5853 | 0.5590 | 33 | 0.5480 | 0.5678 | 0.5312 |
| 16 | 0.5165 | 0.5518 | 0.5312 | 34 | 0.4850 | 0.5092 | 0.4739 |
| 17 | 0.5276 | 0.5686 | 0.5536 | 35 | 0.4981 | 0.5120 | 0.4656 |
| 18 | 0.4838 | 0.5168 | 0.4966 | 36 | 0.5001 | 0.5308 | 0.5003 |
| | | | **Average** | | 0.5443 | **0.5695** | 0.5378 |

On average, the proposed assembled regression with single-token shifting (*w/ shifting*) outperforms the trivial *Dist-1* baseline by 2.52% across all 36 layers, and consistently achieves the highest overlap ratio at most layers. Although AsyncSpade does not dominate every individual layer, the consistent gain at the aggregate level translates into significantly better downstream performance, as further confirmed by the NIAH retrieval results in Section C.

# E. Latency Breakdown on H100 and H200 GPUs

The motivation profiling in Figure 2 is conducted on A100 GPUs. To verify that cache selection remains a significant bottleneck on newer-generation hardware, we further profile Qwen3 models on H200 GPUs with FlashAttention-3.

*Table 6.* Latency breakdown of page-level sparse decoding on H200 with FlashAttention-3, under varied concurrency at fixed 32k context length.

| Model | Batch Size | Total (ms) | Cache Selection (ms) | Cache Selection % |
|-------|-----------|-----------|---------------------|-------------------|
| Qwen3-1.7B | 64 | 7.15 | 3.57 | 49.9% |
| Qwen3-4B | 128 | 15.19 | 5.82 | 38.4% |
| Qwen3-8B | 32 | 5.96 | 2.34 | 39.3% |
| Qwen3-32B | 64 | 13.81 | 4.22 | 30.6% |

*Table 7.* Latency breakdown of page-level sparse decoding on H200 with FlashAttention-3, under varied context length at fixed batch size 32.

| Model | Context Length | Total (ms) | Cache Selection (ms) | Cache Selection % |
|-------|---------------|-----------|---------------------|-------------------|
| Qwen3-1.7B | 128k | 14.04 | 7.08 | 50.4% |
| Qwen3-4B | 128k | 15.37 | 5.84 | 38.0% |
| Qwen3-8B | 64k | 11.58 | 4.60 | 39.7% |
| Qwen3-32B | 64k | 13.75 | 4.23 | 30.8% |

Across all configurations, cache selection accounts for 30–50% of the total decoding latency. The bottleneck observed on A100 (Figure 2) persists on newer-generation GPUs equipped with FlashAttention-3, confirming that the sequential dependency of query-aware sparse decoding is a hardware-agnostic limitation that motivates the asynchronous design of `AsyncSpade`.

