# OpenReview forum: "AsyncSpade: Efficient Test-Time Scaling with Asynchronous Sparse Decoding"
_ICML.cc/2026/Conference — ICML 2026 regular_

### Official Review · Reviewer_WBHU · 2026-03-08

**Soundness:** 4
**Presentation:** 4
**Significance:** 3
**Originality:** 3
**Overall Recommendation:** 5
**Confidence:** 3

**Summary:**

The paper introduces AsyncSpade, a novel framework that decouples KV cache selection from the decoding computation, enabling overlap between the token selection process and the inference computation. Since token selection in sparse attention can become a system bottleneck, AsyncSpade’s asynchronous design significantly improves performance. The framework is comprehensively evaluated to demonstrate the effectiveness of its design.

**Compliance With Llm Reviewing Policy:**

Affirmed.

**Final Justification:**

The submitted draft of this paper is overall above the threshold, elegantly addressing a major problem in sparse attention, i.e., the overlong token selection cost. The idea is good; using disaggregation for sparse attention computation aligns with the current trend that module-level operators should be disaggregated to different devices (see P-D disaggregation, AFD disaggregation, etc.).

During the rebuttal phase, I raised questions regarding motivations, experiments, etc. The authors provide detailed responses. Notably, the authors provide more data on cache selection cost, validating that the core motivation of this paper is correct. The authors also provide more experimental results and more explanations on the flexibility of their system. Thereafter, my major concerns are addressed.

Therefore, in summary, I think most of the weaknesses are addressed, and I prefer to improve the score.

**Key Questions For Authors:**

Q1: Could you provide more evidence that cache selection remains a system bottleneck on H200 GPUs?

Q2: Could you extend Figure 5 to provide additional evidence supporting the effectiveness of your prediction method? Evaluations across more layers and possibly additional models would help better validate the underlying assumption.

Q3: Could you clarify the parallelization strategy used for AsyncSpade and the chosen baselines? In particular, for AsyncSpade, how do you determine the number of GPUs allocated to the Cache Rank versus the Inference Rank? How do you balance the per-step decoding latency with the cache selection computation to achieve efficient overlap?

Q4: I am curious about the concrete KV cache communication overhead. In Table 2 of the current draft, the inference time and cache operation time appear roughly balanced, but how does the KV cache transfer overhead affect the overall performance? Additionally, the paper seems to use a heterogeneous cluster consisting of H100 and A100 GPUs. Does this setup introduce inter-machine communication for AsyncSpade, and if so, how does it impact the bandwidth requirements and overall efficiency?

Q5: Does your method outperform recent state-of-the-art approaches, such as the recently released DeepSeek DSA? A direct comparison would help better position the contribution of this work. I acknowledge it's difficult to present a comprehensive comparison for new baselines in the rebuttal phase; some preliminary results are helpful for convincing the readers.

**Limitations:**

Impact statement is missing.

**Strengths And Weaknesses:**

Strengths:

- This paper addresses a timely and important problem. Token selection can indeed become a system bottleneck in sparse attention inference, and an asynchronous framework provides a natural solution to mitigate this issue.
- The paper demonstrates solid technical depth in the design of the asynchronous execution pipeline. The design of AsyncSpade is well motivated and sound, and the optimization techniques are well reasoned.
- The evaluation is comprehensive, covering both system-level and algorithmic evaluations. The results show significant improvements in system throughput and/or algorithmic convergence compared to the chosen baselines.
- The paper is well written and easy to follow. The presentation is clear, rigorous, and well structured.


Weaknesses:

- The profiling results in Figure 2, which aim to demonstrate that token selection is a system bottleneck, were conducted only on A100 GPUs. However, newer-generation GPUs (e.g., H20 and H200) are now widely available. It remains unclear whether token selection continues to be a system bottleneck on these newer architectures, and additional verification would strengthen the claim.
- The evidence supporting the core motivation of this work—namely, that query token states can be predicted from preceding query tokens—is not sufficiently strong. It would be helpful to extend Figure 5 to include results across more layers and, preferably, additional models.
- The description of the experimental setup introduces some confusion. It is unclear how AsyncSpade and the baselines are deployed on the given hardware.
- More details on the execution pipeline would be helpful, particularly the concrete overhead of transferring the KV cache between the Cache Rank and the Inference Rank.
- The experiments only compare with two baselines. While important baselines, e.g., MoBA, DeepSeek V3.2 (DSA) is missing.

---

> ### Author Rebuttal · Authors · 2026-03-31
>
> Dear reviewer WBHU,
>
> We sincerely thank your constructive feedback and address your questions as below.
> ### 1. H200 breakdown
> We provide H200 profilings with FlashAttention-3, where cache selection remains a bottleneck in query-aware sparse decoding.
> #### Varied Concurrency (Context Length 32k)
> |Scale|Batch Size|Total (ms)|Cache Selection (ms)|Attention Core (ms)|Cache Selection %|
> |:-:|:-:|:-:|:-:|:-:|:-:|
> |1.7B|64|7.15|3.57|3.43|49.9%|
> |4B|128|15.19|5.82|9.11|38.4%|
> |8B|32|5.96|2.34|3.42|39.3%|
> |32B|64|13.81|4.22|9.11|30.6%|
>
> #### Varied Context Length (Batch Size 32)
> |Scale|Context Length|Total (ms)|Cache Selection (ms)|Cache Selection %|
> |:-:|:-:|:-:|:-:|:-:|
> |1.7B|128k|14.04|7.08|50.4%|
> |4B|128k|15.37|5.84|38.0%|
> |8B|64k|11.58|4.60|39.7%|
> |32B|64k|13.75|4.23|30.8%|
>
> Cache selection consumes a large proportion of total decoding latency across varied scales and configurations.
> ### 2. Comparison with DSA/MoBA
> We respectfully clarify that DSA/MoBA address different problem dimensions compared to AsyncSpade:
> - **Distinction**: DSA/MoBA are post-training architectural modifications requiring retraining / fine-tuning to learn sparse patterns (block-wise routing in MoBA, learned sparsity in DSA). AsyncSpade is a training-free decoding acceleration framework.
> - **Target**: DSA/MoBA primarily target 128k+ long-context understanding, whereas AsyncSpade specifically optimizes Test-Time Scaling serving, where the bottleneck is the heavy decoding overhead.
> - **Relationship**: These approaches are orthogonal and synergistic. AsyncSpade's asynchronous framework can be applied on top of DSA/MoBA-tuned models, where *Cache Rank* can perform token selection within the sparse blocks identified by MoBA's router.
> ### 3. KV Cache Communication & Hardware Configuration
> **Hardware Setup**: We do not use a heterogeneous cluster mixing A100 and H100. Instead, we conduct experiments on two separate, homogeneous nodes: N1 (8× A100) and N2 (8× H100) (Table 1). All reported results use identical GPUs in the same node without inter-machine communication between A100 and H100.
>
> **Overlap Feasibility**: As shown in Figure 2, *Cache Selection* constitutes 30-50% of total latency in Quest. AsyncSpade disaggregates this to a separate rank for overlapping. We observe that **full overlap is achievable within a certain workload range**, where *Cache Rank* process remains shorter than the forward pass of *Inference Rank*. Beyond this range, KV selection overhead grows with batch size and context length, exceeding the forward computation window and limiting the achievable speedup, a trade-off to the disaggregated design.
>
> We will clarify in Section 5.1, and add explicit quantification of communication volume vs. computation time in Appendix.
> ### 4. Parallelization Strategy and GPU Allocation
> The split between *Inference Rank* and *Cache Rank* is configurable based on workload rather than fixed. The general principle is the *Inference Rank* hosts the layer partitions of the model layers, while the *Cache Rank* requires sufficient resources to complete (1) temporal-regressive query prediction, (2) token-level top-k selection, and (3) KV cache re-organization within the time window of the *Inference Rank*'s forward computation. We acknowledge that the optimal GPU allocation ratio depends on specific deployment scenarios. We will add a sensitivity analysis of different Inference/Cache Rank configurations in the revised version.
> ### 5. Layer-wise Validation & Downstream Verification
> To extend Figure 5, we provide layer-wise overlap ratio analysis for Qwen3-8B (36 layers) and Needle-in-a-Haystack (NIAH) test.
> #### Overlap Ratio
> We compare with Distance-1 (previous-query) for all layers:
> |Layer|Dist-1|AsyncSpade|
> |:-:|:-:|:-:|
> |1|0.5266|0.5681|
> |2|0.5511|0.5866|
> |3|0.5892|0.6240|
> |4|0.5878|0.6151|
> |5|0.5672|0.5922|
> |6|0.5891|0.6096|
> |7|0.5290|0.5551|
> |8|0.5524|0.5578|
> |9|0.5665|0.5839|
> |10|0.4863|0.5140|
> |11|0.6300|0.6562|
> |12|0.5574|0.5904|
> |13|0.4876|0.5181|
> |14|0.4824|0.5083|
> |15|0.5570|0.5853|
> |16|0.5165|0.5518|
> |17|0.5276|0.5686|
> |18|0.4838|0.5168|
> |19|0.5118|0.5456|
> |20|0.5453|0.5752|
> |21|0.5174|0.5405|
> |22|0.5429|0.5584|
> |23|0.5644|0.5849|
> |24|0.6092|0.6226|
> |25|0.5473|0.5544|
> |26|0.5890|0.6043|
> |27|0.5351|0.5480|
> |28|0.5487|0.5685|
> |29|0.5673|0.5781|
> |30|0.5176|0.5382|
> |31|0.5930|0.6184|
> |32|0.5536|0.5764|
> |33|0.5480|0.5678|
> |34|0.4850|0.5092|
> |35|0.4981|0.5120|
> |36|0.5001|0.5308|
> |Avg|0.5412|0.5664|
>
> AsyncSpade outperforms at 32 out of 36 layers.
> #### Downstream Validation via NIAH
> While the average overlap improvement appears modest (+2.52%), we demonstrate via crafted hard NIAH test that this translates to **disproportionately large gains** in long-context retrieval:
> |Context Length|Full Attention|AsyncSpade|Dist-1|
> |:-:|:-:|:-:|:-:|
> |10k|0.99|0.93|0.74|
> |20k|0.98|0.97|0.77|
>
> We will provide complete layer-wise analysis and NIAH test in Appendix to demonstrate the correlation between overlap ratio proxy and downstream task performance.

---

> > ### Author Rebuttal · Reviewer_WBHU · 2026-04-02
> >
> > Thank you for the detailed rebuttal. The authors have addressed my main concerns. Overall, the rebuttal has improved my assessment and understanding of the paper, and I will raise my score.

---

> > > ### Author Response · Authors · 2026-04-02
> > >
> > > We sincerely thank Reviewer WBHU for the careful consideration of our rebuttal and the positive feedback. We are delighted that our rebuttal adequately addressed your concerns. We will incorporate all the discussed improvements (H200 profiling details, layer-wise overlap analysis, NIAH validation, and hardware configuration clarifications) into the final version of the paper.

---

### Official Review · Reviewer_G9xv · 2026-03-12

**Soundness:** 3
**Presentation:** 2
**Significance:** 4
**Originality:** 3
**Overall Recommendation:** 4
**Confidence:** 3

**Summary:**

The authors present AsyncSpade, a framework for sparse attention serving that decouples sparse KV-cache selection from the main LLM inference pipeline in order to hide the latency overhead. This is realized through a dual-rank design where the inference rank performs the LLM forward pass, while a cache rank on a separate GPU predicts and fetches selected KV cache entries at token granularity.

The key mechanism is a query vector predictor implemented as a linear regression model over the queries of previous tokens. The authors claim that this design enables complete overlap between KV-cache selection and the rest of the inference computation, while maintaining high prediction accuracy relative to full dense attention. As a result, AsyncSpade provides an efficient sparse attention implementation that scales better to long-context scenarios while maintaining accuracy comparable to full attention.

**Compliance With Llm Reviewing Policy:**

Affirmed.

**Final Justification:**

I maintained my weak accept score for this paper after the author's rebuttal, which reinforced my initial assessment.

**Key Questions For Authors:**

1. In Figure 5, the overlap ratio does not appear substantially higher than the overlap ratios shown in Figure 4 (particularly the distance-1 curve). Could you clarify why the regression-based prediction is expected to improve token selection beyond simply using nearby queries?

2. The overlap ratio is used as a proxy for prediction quality, but the reported values suggest that the predicted selection is not perfect. Given that downstream task accuracy approaches full attention, what factors compensate for tokens that are missed? Can you quantify the quality of token selection relative to baselines?

3. The paper introduces the *assembled regression* approach, but its effect on prediction quality is not clearly shown. Could you provide quantitative comparisons showing how much improvement this method provides relative to simpler regression approaches?

4. Figure 10 reports end-to-end TPOT improvements. A latency breakdown similar to Figure 2 for all methods shown in Figure 10 would help better understand where these gains originate.

5. The paper frequently refers to achieving "theoretically optimal TPOT." Could you clarify what notion of optimality is intended here? The argument appears to rely on a fully overlapped execution model rather than a formal lower bound, so clarifying this claim would help.

**Strengths And Weaknesses:**

### Strengths

- novel query prediction approach for sparse attention with promising accuracy. The use of temporal locality in query embeddings to predict the next query state is a simple but effective idea that enables asynchronous KV selection.
- end-to-end engineering of the inference engine. The system design that decouples KV cache selection from the inference pipeline and overlaps it with computation is well motivated and demonstrates careful systems design.
- good benchmark performance at lower compute and inference latency compared to strong baselines. The evaluation across multiple reasoning benchmarks and models shows consistent TPOT improvements while maintaining accuracy comparable to full attention.

### Weaknesses

#### Clarity and presentation

- in introduction, *attention core* should be defined before being used. It is possible to infer that it denotes the attention stage after the sparse KV selection has taken place, but it requires some reading around.
- also in introduction, when the two special ranks are introduced, it is unclear what *ranks* are conceptually within this paragraph. A revision in terminology or even a brief explanation (e.g., *logical module / worker*) could work.
- in figure 3, defining *Cache selection*, *Attention Core*, and *Model Parameters* would be beneficial. It can be inferred, but clear definitions would improve clarity.
- in equation (3), we define \( \omega^* \) in the equation but not in the text. An explanation should clarify what this represents. Also in the sentence in the same section (3.2), when you say *"we also apply \( \omega^* \) on inputs"*, it is unclear what *inputs* refers to.
- in section 4.1, it feels odd to call the KV selection operation *redundant*. Perhaps phrasing like *moving the KV selection operation off the critical path* would work better.
- in section 4.2, you say *"by solving the ridge regression problem in Equation (3), also with the softmax-normalization in Equation (4)"*, but it is unclear whether \( \omega \) refers to the weights from solving equation (3) or the weights after normalization.
- in the same section, the phrase *"can even surpass that of the counterpart without shifting"* is not intuitive and could benefit from explanation.
- in the same section, when talking about *assembled regression*, terms like *aggregated* or *ensemble* may convey the meaning more clearly.
- minor terminology nitpick: the term *"duo-rank"* is somewhat unusual. A term like *"dual-rank"* or simply *"two-rank architecture"* may be clearer.

#### Missing analysis or clarification

- the predicting power or *overlap ratio* of this assembled regression should be shown to clearly demonstrate its effect, as well as a small study on which lengths of tokens are usually selected or a distribution over them.
- the overlap ratio in figure 5 does not appear much higher than the overlap ratio in figure 4, particularly the distance-1 plot. Clarification would help explain this and whether something is being misinterpreted.
- overlap ratio is a good proxy but given the numbers across layers it does not seem perfect. Since downstream task accuracies approach full attention, it would help to explain what compensates for the tokens that are missed. It would also be useful to quantify how good of a token selection AsyncSpade performs relative to baselines, either using overlap ratio or another metric.
- a latency breakdown similar to figure 2 for all methods in figure 10 would help better explain the performance differences observed there.

#### Overall Assessment

Overall, the paper proposes an interesting systems approach to improving sparse decoding efficiency and demonstrates promising empirical results. While the idea of asynchronous KV selection and the associated engineering are compelling, several aspects of the presentation and analysis could be clarified, particularly around the query prediction mechanism and the interpretation of the overlap ratio results. With clearer explanations and additional analysis, the work could be a strong contribution to efficient LLM inference systems.

Many of the weaknesses identified are also linked to questions in the 'Key Questions for Authors' section below.

---

> ### Author Rebuttal · Authors · 2026-03-31
>
> Dear reviewer G9xv,
>
> We sincerely thank your constructive feedback, and address your questions step by step.
> ### 1. Evaluation proxy
> In this paper, we use the overlap ratio for selected tokens as the proxy and omitted to build the relation between it and the model performance. We provide supplementary experiments using Needle in a Haystack (NIAH) test. We build a hard examination by inserting a single short needle embeds a 20-digit random code associated with a randomly sampled city, phrased in a low-salient way (“reference code …”) rather than with an explicit memorization cue. We place the needle at 50 evenly spaced relative depths, and run for 2 rounds. We examine Qwen3-32B model at varied context length using the methods mentioned in our paper, and use 1/8 sparsity for the sparse decoding methods.
>
> 10k Context Length:
> |full attention|AsyncSpade|Distance-1|Distance-2|Distance-3|Distance-4|
> |:-:|:-:|:-:|:-:|:-:|:-:|
> |0.99|0.93|0.74|0.56|0.43|0.48|
>
> 20k Context Length:
> |full attention|AsyncSpade|Distance-1|Distance-2|Distance-3|Distance-4|
> |:-:|:-:|:-:|:-:|:-:|:-:|
> |0.98|0.97|0.77|0.77|0.61|0.64|
>
> The NIAH evaluations demonstrate that AsyncSpade can basically preserve the long context retrieval capability of full attention and significantly outperforms trivial approaches such as filtering KV cache with only the previous query token. This is consistent with the proxy (overlap ratio) used in our paper, where AsyncSpade > Distance-1 > Distance-2 > Distance-3 > Distance-4.
>
> ### 2. Overlap ratio comparison
> Limited to the space, we did not provide a comprehensive investigation on the overlap ratio for each layer. Our primary goal is building a general approach to better preserve the capability of the original model, and although AsyncSpade cannot outperform at each layer, it can deliver higher overlap ratio at most layers, and demonstrate significantly better performance as shown in the NIAH test. We list the average overlap ratio for each layer of DeepSeek-R1-Qwen3-8B as below.
> |Layer|Dist-1|w/ shifting|w/o shifting|
> |:-:|:-:|:-:|:-:|
> |1|0.5266|0.5681|0.5483|
> |2|0.5511|0.5866|0.5608|
> |3|0.5892|0.6240|0.6025|
> |4|0.5878|0.6151|0.5844|
> |5|0.5672|0.5922|0.5622|
> |6|0.5891|0.6096|0.5716|
> |7|0.5290|0.5551|0.5223|
> |8|0.5524|0.5578|0.5097|
> |9|0.5665|0.5839|0.5457|
> |10|0.4863|0.5140|0.4800|
> |11|0.6300|0.6562|0.6300|
> |12|0.5574|0.5904|0.5657|
> |13|0.4876|0.5181|0.4852|
> |14|0.4824|0.5083|0.4739|
> |15|0.5570|0.5853|0.5590|
> |16|0.5165|0.5518|0.5312|
> |17|0.5276|0.5686|0.5536|
> |18|0.4838|0.5168|0.4966|
> |19|0.5118|0.5456|0.5296|
> |20|0.5453|0.5752|0.5517|
> |21|0.5174|0.5405|0.5141|
> |22|0.5429|0.5584|0.5281|
> |23|0.5644|0.5849|0.5556|
> |24|0.6092|0.6226|0.5961|
> |25|0.5473|0.5544|0.5194|
> |26|0.5890|0.6043|0.5717|
> |27|0.5351|0.5480|0.5140|
> |28|0.5487|0.5685|0.5372|
> |29|0.5673|0.5781|0.5389|
> |30|0.5176|0.5382|0.5057|
> |31|0.5930|0.6184|0.5919|
> |32|0.5536|0.5764|0.5455|
> |33|0.5480|0.5678|0.5312|
> |34|0.4850|0.5092|0.4739|
> |35|0.4981|0.5120|0.4656|
> |36|0.5001|0.5308|0.5003|
>
> In average, AsyncSpade outperforms the trivial method (Distance-1, filtering the KV cache with previous query) with 2.52%, and in the NIAH test, AsyncSpade delivers similar performance compared to full attention, significantly outperforms the trivial method.
>
> ### 3. Theoretically optimal TPOT
> Thank you for this insightful comment. We clarify that the "optimality" refers to the theoretically minimum latency for query-aware sparse decoding, where the overhead of token selection is entirely eliminated from the inference pipeline.
> - **Definition of Optimality**: In existing query-aware sparse decoding (e.g., Quest), KV cache selection is sequentially dependent on the current query, adding non-trivial latency to each decoding step. We define "theoretically optimal TPOT" as achieving a TPOT equal to the execution of the model’s forward pass (inference pipeline) alone, as if the sparse selection mechanism incurred zero time cost.
> - **Realization via Asynchronous Overlap**: AsyncSpade achieves this by decoupling KV management into a dedicated Cache Rank. By using a temporal-regressive module to predict the next query, we pre-select and transfer the KV cache for the next token while the current token is still being processed in the Inference Rank.
> - **Condition for Optimality**: As long as the latency of (1) cross-device P2P communication and (2) Cache Rank operations (criticality estimation, top-k selection, and memory reorganization) remains lower than the forward computing latency on the Inference Rank, these overheads are fully "hidden". In this state, the system reaches its theoretical efficiency limit for sparse decoding.
>
> We will update the manuscript to explicitly define this notion of optimality and clarify that it pertains to the elimination of sequential bottlenecks within the sparse decoding framework.
>
> ### 4. Clarity & Figure 10
> We sincerely appreciate your careful review and helpful suggestions, and will fix these issues and update Figure 10 in our revised version.

---

> > ### Author Rebuttal · Reviewer_G9xv · 2026-04-04
> >
> > My questions have been adequately answered. Happy to maintain my positive-leaning score.

---

### Official Review · Reviewer_uKaK · 2026-03-13

**Soundness:** 3
**Presentation:** 2
**Significance:** 3
**Originality:** 3
**Overall Recommendation:** 3
**Confidence:** 1

**Summary:**

This paper proposes an asynchronous sparse decoding framework named AsyncSpade, which completely decouples the precise selection of the KV Cache from the model's forward inference process and executes them in parallel by predicting the next token's query state in advance. This innovation breaks the sequential computation bottleneck of existing query-aware sparse decoding, achieving fine-grained, token-level cache selection without sacrificing the large model's reasoning accuracy, and significantly reducing the TPOT for long-context generation in TTS.

**Compliance With Llm Reviewing Policy:**

Affirmed.

**Key Questions For Authors:**

see weaknesses.

**Limitations:**

See weaknesses.

**Strengths And Weaknesses:**

### Strengths
1. Revealing query state patterns: Directly addresses the sequential bottleneck in existing sparse decoding by empirically validating the strong temporal locality and linear correlation among adjacent query states.
2. Training-free, lightweight prediction: Based on these observations, the paper proposes using a multi-window ridge regression to predict the next query in advance; this approach incurs extremely low computational overhead, is completely training-free, and is highly generalizable.
3. Dual-architecture breaking the sequential bottleneck: Pioneers a decoupled dual-architecture comprising an Inference Rank and a Cache Rank, which completely eliminates the massive latency caused by waiting for query computation seen in traditional methods.
4. Efficient algorithm-system co-design: Incorporates low-level hardware implementation optimizations, such as introducing "Depth-wise Parallelism" to pack and transmit the states of multiple layers, thereby maximizing inter-node bandwidth utilization.

### Weaknesses
1. Strong reliance on multi-GPU deployment, lacking general applicability: Its core dual-rank architecture, consisting of an Inference Rank and a Cache Rank, strictly mandates a multi-GPU environment (such as the 8xA100 or 8xH100 setups used in the experiments). This makes the framework completely inapplicable to single-GPU systems, edge devices, or resource-constrained deployment scenarios.
2. Extremely stringent requirements on GPU interconnect bandwidth: The perfect asynchronous acceleration heavily relies on high-speed inter-node communication to successfully mask transmission latency. According to the data in Table 2, achieving full communication-computation overlap requires an extremely high minimal bandwidth ranging from approximately 105 GB/s to 230 GB/s. In a standard PCIe environment, the cross-node transmission latency would directly negate the speed benefits introduced by the asynchronous design.
3. Insufficient validation of task generalization: The paper's experiments are strictly limited to mathematical and logical reasoning tasks that inherently possess strong coherence. The authors have not validated whether the prediction algorithm, which is based on a local sliding window of recent queries, would cause severe performance degradation in scenarios with highly dispersed attention or massive contextual spans (e.g., "Needle In A Haystack" tasks or multi-document QA).

As an additional point, the abstract is somewhat lengthy and would benefit from being more concise. And a large blank space in the bottom right corner of page 7. Adjusting the layout would make it look much better.

---

> ### Author Rebuttal · Authors · 2026-03-30
>
> Dear reviewer uKaK,
>
> We sincerely appreciate your thoughtful review and constructive feedback. We are especially grateful that you highlighted the strengths of our work, including **the empirical observation of query-state regularity, the lightweight training-free prediction design, the decoupled asynchronous architecture, and the effective hardware-aware system optimization.**
>
> ---
>
> **W #1. Reliance on multi-GPU deployment**
>
> We acknowledge that AsyncSpade's disaggregated design is targeted at distributed data-center serving for long-CoT test-time scaling, rather than single-GPU or edge deployment. We respectfully argue that **this is a deliberate systems choice rather than a limitation for our target use case**. As noted in Section 1 (lines 66-71), test-time scaling with extended chain-of-thought reasoning inherently requires large KV cache memory footprints and high serving concurrency that exceed single-GPU capacity (e.g., 32B models decoding 32k+ tokens).
>
> The parameter computation during inference (regular MatMul or Grouped GeMM for MoE) and fine-grained token-level KV selection (irregular memory accesses with dynamic indexing) constitute heterogeneous workloads with conflicting optimization objectives. **Efficiently overlapping these on a single GPU would require complex warp-level scheduling or persistent kernel designs that significantly increase implementation complexity and compromise kernel efficiency**.
>
> We agree that this restricts AsyncSpade's applicability to resource-constrained environments, and we will clarify this scope explicitly in the introduction. We also note that the **core algorithmic insight, i.e., temporal-regressive query prediction, is hardware-agnostic and can be applied to single-GPU deployments through CUDA streams or CPU offloading**.
>
> Finally, we emphasize that **AsyncSpade is orthogonal to KV cache compression techniques** (e.g., 4-bit KV cache quantization). While AsyncSpade optimizes runtime orchestration, quantization can reduce the bandwidth requirement by reducing per-token storage footprint. We defer the systematic evaluation of such hybrid configurations to our future work.
>
> ---
>
> **W #2. Stringent requirements on GPU interconnect bandwidth**
>
> We agree that achieving fully hidden communication requires high-bandwidth intra-node P2P interconnects (e.g., NVLink 250-350 GB/s as in Table 1). However, we respectfully clarify that **AsyncSpade does not transmit the full KV cache during steady-state decoding, and it can still provide benefits even without perfect overlap.**
>
> **Communication volume**: After initialization, each decoding step only transfers: (1) packed current q/k/v states for the window (negligible volume: batch_size * num_layers * num_heads * window_size), and (2) the selected sparse KV entries for the next step, rather than the full context KV cache. (e.g., 2k tokens out of 32k total context, as configured in Section 5.2). This reduces per-step communication volume proportionally to the sparsity ratio.
>
> **Partial Overlap still helps**: While our current evaluation uses NVLink-equipped nodes (A100/H100), the asynchronous framework inherently decouples KV selection latency from the critical path. Even with constrained bandwidth (e.g., PCIe-based interconnects), AsyncSpade eliminates the sequential dependency bottleneck that plagues query-aware methods like Quest (Figure 2), where cache selection blocks the entire forward pass. The precise TPOT improvement under lower bandwidth depends on the communication-computation ratio and requires systematic evaluation, which we will add to Appendix.
>
> **Future Direction**: Communication optimization is an important direction of our future work. Techniques such as transfer quantization or coarser-grained selection could further reduce bandwidth demand. **These optimizations are compatible with AsyncSpade's framework but were excluded due to space constraints**.
>
> ---
>
> **W #3. Further Validation of Prediction Algorithm**
>
> We run additional **needle-in-a-haystack (NIAH)** experiments on the two models used in our main study, each evaluated based on the maximum practical context length. **AsyncSpade shows no performance degradation in NIAH experiments across all tested context lengths on both models.**
>
> The haystack is built from contiguous PG19-style book text from a JSONL corpus. We build a hard examination by inserting a single short needle that embeds a 10-digit random code associated with a randomly sampled city, phrased in a low-salient way (“reference code …”) rather than with an explicit memorization cue. We place the needle at 20 evenly spaced relative depths (including the end), snapping non-terminal insertions to the preceding sentence boundary. Results are listed below.
>
> |DeepSeek-R1-Qwen3-8B|16k|32k|64k|128k|
> |:-:|:-:|:-:|:-:|:-:|
> |w/o AsyncSpade|1|1|1|1|
> |w/ AsyncSpade|1|1|1|1|
>
> |Qwen3-32B|10k|20k|30k|40k|
> |:-:|:-:|:-:|:-:|:-:|
> |w/o AsyncSpade|1|1|1|1|
> |w/ AsyncSpade|1|1|1|1|

---

### Decision · Program_Chairs · 2026-04-30

**Decision:**

Accept (regular)

**Comment:**

The paper proposes AsyncSpade, an asynchronous sparse decoding framework that predicts the next query state from recent queries and decouples KV-cache selection from the main autoregressive inference pipeline to remove the sequential dependency that limits prior query-aware sparse decoding methods and enable finer-grained token-level selection.

The reviewers generally agree that the paper is technically strong and addresses a meaningful problem. Its main strengths are the clear motivation, the novel asynchronous design, the regressive prediction method, and the strong empirical results showing improved TPOT while maintaining competitive reasoning accuracy across multiple benchmarks.

The main concerns raised during review focused on the scope of applicability, especially the reliance on multi-GPU deployment and high-bandwidth interconnects, as well as the need for additional validation beyond reasoning tasks. In the discussion and rebuttal, the authors clarified that the method is intentionally designed for distributed data-center inference rather than single-GPU or edge deployment, provided additional explanation of the communication pattern and overlap assumptions, and added further validation including needle-in-a-haystack experiments. These responses addressed most of the reviewers’ concerns. All reviewers who gave acknowledgement mentioned that their concerns are fully addressed.

Overall, I find the paper to make a novel and practically meaningful contribution to efficient LLM inference. I therefore recommend acceptance. The authors should use the reviewers' suggestions to revise the paper and incorporate promised modifications and clarifications.